# Formulation and Characterization of Polymeric Cross-Linked Hydrogel Patches for Topical Delivery of Antibiotic for Healing Wound Infections

**DOI:** 10.3390/polym15071652

**Published:** 2023-03-27

**Authors:** Sana Afzal, Kashif Barkat, Muhammad Umer Ashraf, Ikrima Khalid, Yasir Mehmood, Nisar Hussain Shah, Syed Faisal Badshah, Saba Naeem, Saeed Ahmad Khan, Mohsin Kazi

**Affiliations:** 1Faculty of Pharmacy, University of Lahore, Lahore 54590, Pakistan; 2Department of Pharmaceutics, Faculty of Pharmaceutical Sciences, Government College University Faisalabad, Faisalabad 38000, Pakistan; 3Department of Pharmacy, Kohat University of Science and Technology, Kohat 26000, Pakistan; 4Division of Molecular Pharmaceutics and Drug Delivery, College of Pharmacy, The University of Texas at Austin, Austin, TX 78712, USA; 5Department of Pharmaceutics, College of Pharmacy, King Saud University, P.O. Box 2457, Riyadh 11451, Saudi Arabia

**Keywords:** hydrogel, topical drug delivery, Franz diffusion cell, skin irritation study, hydrogel patches, bacitracin zinc

## Abstract

Wound healing faces significant challenges in clinical settings. It often contains a series of dynamic and complex physiological healing processes. Instead of creams, ointments and solutions, alternative treatment approaches are needed. The main objective of the study was to formulate bacitracin zinc-loaded topical patches as a new therapeutic agent for potential wound healing. A free radical polymerization technique was optimized for synthesis. Polyethylene glycol-8000 (PEG-8000) was chemically cross-linked with acrylic acid in aqueous medium, using Carbopol 934 as a permeation enhancer and tween 80 as surfactant. Ammonium persulfate and N,N’-Methylenebisacrylamide (MBA) were utilized as initiator and cross-linker. FTIR, DSC, TGA, and SEM were performed, and patches were evaluated for swelling dynamics, sol-gel analysis, in vitro drug release in various media. A Franz diffusion cell was used for the permeation study. Irritation and wound healing with the drug-loaded patches were also studied. The characterization studies confirmed the formation of a cross-linked hydrogel network. The highest swelling and drug release were observed in formulations containing highest Polyethylene glycol-8000 and lowest N,N’-Methylenebisacrylamide concentrations. The pH-sensitive behavior of patches was also confirmed as more swelling, drug release and drug permeation across skin were observed at pH 7.4. Fabricated patches showed no sign of irritation or erythema as evaluated by the Draize scale. Faster wound healing was also observed with fabricated patches compared to marketed formulations. Therefore, such a polymeric network can be a promising technology for speeding up wound healing and minor skin injuries through enhanced drug deposition.

## 1. Introduction

Skin is a large and complex multilayer system that functions as the interface between the human body and the external environment. The primary defense function of the skin makes it a more effective barrier against many harmful environmental threats to the life it encloses [1]. Wounds occur due to minor or major cuts, destroying the normal physiology and continuity of the skin’s epithelial mucosa [2]. Wound healing is a dynamic and coordinated process of growth and tissue regeneration that requires the proper physiological environment and conditions for its treatment, otherwise it will lead to the colonization of infectious agents in the wound [3]. Since the wound exudate does not respond efficiently to the topical application of the antibiotics, the methodology adopted to deliver the drug is of vast importance, as it not only enhances the efficacy of the therapeutic agent but also improves its pharmacological and therapeutic response [1].

Traditional drug delivery systems are associated with a number of limitations, such as toxicity of the vital organs and frequent dosing as compared to that of hydrogels [4]. Since the traditional drug administration often involves high dosing and frequent administration to promote therapeutic effects [5], it may therefore result in compromised efficacy along with severe side effects [6]. Hydrogels have played a vital role in minimizing the drawbacks associated with conventional drug delivery systems [7]. Recently, great efforts have been made by researchers to design wound dressings that stay in contact with the wound, not only to protect the wound from the bacterial growth but also to provide proper conditions, such as a moist environment, allowing gaseous exchange between the wound and the environment and tissue regeneration, resulting in an enhanced healing process [8]. Hydrogel dressings promote moist healing and are suitable for cleaning dry, sloughy and necrotic wounds. Hydrogel networks are hydrophilic in nature and are produced either by physical or chemical cross linking of their macromolecular, complex and three-dimensional polymeric networks. The swelling capability of hydrogels in an aqueous medium provides rate-controlling barriers for the diffusion of water-soluble drugs, allowing the highest attainable fluxes from polymers. Chemical cross-linking is responsible for the network structure and physical integrity [9,10,11].

Topical patches with hydrogel matrices are a new advancement in the drug delivery technique that enhances the bioavailability of the therapeutic agent at the infection site and in the blood stream. They absorb the wound exudate by releasing the drug at the infection site, speeding up the healing process by promoting cell adhesion, cell proliferation and direct cell migration. This painless and non-invasive technique deliver the drug across the skin to the circulation and maintain the therapeutic concentration in a controlled manner [12].

PEG-8000 is considered as a perfect candidate for biological formulations due to its ideal properties, such as non-toxicity, biocompatibility, flexibility, and its role in the controlled release of drug [13]. Novel hydrogel topical patches prepared from PEG provide controlled release of drugs. They have improved film-forming characteristics, enhanced mechanical strength, and adequate adhesion to soft tissues. Acrylic acid (AA) is a hydrophilic monomer which is extensively utilized in the formation of pH-sensitive hydrogels due to the presence of the carboxylic acid group, which undergoes ionization at higher pH, thus imparting pH-sensitive behavior to hydrogels. Bacitracin zinc is a renowned broad-spectrum antibacterial agent that acts against gram-positive bacterial species (*cocci* and *bacilli*, including *staphylococcus aureus* and *Streptococci*) and some species of gram-negative bacteria as well [14,15]. The mechanism by which it kills bacteria is by interrupting the synthesis of peptidoglycan, thus leading to the bacterial cell wall disruption. It is a topical antibiotic that is prescribed for the prevention and treatment of wound infections caused by scrapes, small cuts, burns, and aftercare of procedures such as tattoos and circumcision. Orally administered bacitracin zinc has negligible absorption from the gastrointestinal tract [14], whereas its intramuscular and intravenous administration causes nephrotoxicity. Therefore, the topical route is a suitable resort for bacitracin administration for its antibacterial action on wounds. Bacitracin is available in ointment form in the market, which has a slow healing capability of wounds and often takes a long time in the healing process. In comparison to marketed formulations, formulated hydrogel patches of the same drug have shown significant results in wound healing. Its controlled-release drug pattern has improved patient compliance along with its enhanced healing efficiency. The purpose of this research was to synthesize a polymeric cross-linked hydrogel topical patch containing antibiotic bacitracin by means of free radical polymerization.

## 2. Materials and Methods

### 2.1. Materials

Bacitracin zinc was purchased from Wilshire Laboratories, Lahore, Pakistan. Polyethylene glycol-8000 (PEG-8000) was purchased from Merk, Darmstadt, Germany. Acrylic acid (AA) from Daejung ltd. Carbopol 934 from Sigma-Aldrich, Schnelldorf, Germany. Ammonium persulfate (APS) from Merck LGea, Darmstadt, Germany. N, N’-Methylenebisacrylamide (MBA) from Sigma-Aldrich, Schnelldorf, Germany. Tween 80 was obtained from Hoover Pharmaceuticals, Lahore, Pakistan. Sodium hydroxide (NaOH), Potassium dihydrogen phosphate (KH_2_PO_4_), Hydrochloric acid (HCL) and Methanol were procured from Merck, Darmstadt, Germany. Distilled water from Research Lab, University of Lahore, Lahore, Pakistan.

### 2.2. Fabrication of Polymeric Cross-Linked Hydrogel Patches

Free radical polymerization technique was optimized for the formulation of polymeric cross-linked hydrogel patches of bacitracin zinc [16]. Patches were fabricated using PEG-8000 as polymer, Carbopol 934 as a penetration enhancer, and Tween 80 as a surfactant. Separate solutions of the weighed amount of polymer, penetration enhancer, surfactant, APS as initiator, and MBA as cross-linker were prepared at specific temperature. AA was weighed in a separate beaker in the mentioned quantity. The Carbapol 934 solution, Tween solution, monomer, initiator, and cross-linker solutions were added slowly (drop wise) in polymer solution [13]. The final mixture was transferred into labeled petri dishes, which were covered with aluminum foil and placed in preheated water bath at 50 °C for 3 h and then at 60 °C for 24 h. After the specified time, labeled petri dishes were taken out from the water bath. Formulated patches were removed from the petri dishes and were washed with water–ethanol solution in the ratio of 70:30 to remove the impurities and the non-reactive constituents. Patches were placed in the oven at 40 °C for 2 days for complete drying. Dried polymeric patches were then stored in airtight containers for further tests and characterizations. A total number of nine (09) formulations were prepared which were given the code PEGAAM for identification with specific numbers. The composition of fabricated PEGAAM formulations is given in Table 1.

## 3. Characterization of the Fabricated Polymeric Cross-Linked Patches

### 3.1. Fourier-Transform Infrared Spectroscopy (FTIR Spectroscopy)

FTIR spectroscopy of the drug, polymer, permeation enhancer, surfactant, monomer, and cross-linker was performed to determine the interaction among different moieties. The principle behind this study is that the chemical bonds of a substance absorb infrared light at specific frequency and become excited. Samples were completely crushed to fine particle size and were applied on the crystal spot with the help of a spatula for recording the respective absorption spectra. Spectra were recorded in the range of 4000–400 cm^−1^.

### 3.2. Scanning Electron Microscopy (SEM)

SEM provides a clear image of the morphological features or structural analysis of the network of hydrogel topical patches. Hydrogel patches were dried completely and were crushed. Sample of dried crushed patches was placed on the aluminum stab coated with gold. A beam of electrons was focused on the sample. The obtained images showed detailed information regarding the structural morphology of the patch.

### 3.3. Powder X-ray Diffraction (PXRD)

PXRD provides detailed information regarding the unit cell dimension of crystalline or amorphous materials. The crystallinity of the cross-linked hydrogel topical patch was also assessed through this technique. The sample was completely ground to fine particle size, homogenized and was placed in a plastic sample holder of the instrument. Diffraction pattern through the sample material was recorded and analyzed.

### 3.4. Thermogravimetric Analysis (TGA)

Thermogravimetric analysis (TGA) is a thermal analysis which is performed to determine the thermal stability, degradation stabilities, and reaction kinetics of materials that experience mass gain or loss due to the decomposition reaction, oxidation reaction or loss of moisture. An optimized unloaded formulation was minced completely and passed through sieve 40 to obtain the desired size. Samples ranging from 0.5 to 5 mg were placed in a platinum pan and were heated from 25 to 400 °C at a rate of 20 °C/min under purged nitrogen. The analysis was performed, and the average results were recorded.

### 3.5. Differential Scanning Calorimetry (DSC)

In order to analyze the transition temperature of the optimized formulation of the topical patch, DSC was performed. Samples ranging from 0.5 to 3 mg were weighed and placed in an aluminum pan. Samples were processed in the presence of nitrogen gas at a temperature range of 20 to 600 °C and heating was maintained at a temperature of 20 °C/min. Analysis was performed and results were recorded.

### 3.6. Swelling Study

Swelling behavior of formulated hydrogel patches was evaluated using three different phosphate buffers of pH 5.5, 6.5, and 7.4. Formulated polymeric hydrogel patches were first weighed and then soaked in respective buffer solution at room temperature for 36 h. At different time intervals, readings were taken by removing the patches from their respective buffer solutions. Excessive water was removed by cleaning with blotting paper and weighed again on an analytical weighing balance and then the patches were immersed again in respective buffer solutions. Study was conducted until constant weight of the patches was obtained. Swelling ratio termed as (*S*) was obtained from the mentioned equation; (*Ws*) is the swollen hydrogel weight, whereas (*Wd*) is the dried hydrogel weight [16].
(1)$$S=\frac{W_{\;s}-W_{d\;}}{W{\;}_{d}}\times 100$$

### 3.7. Sol-Gel Analysis of Topical Patch

By this technique, the fraction of polymer that remained uncross-linked in the formulation of hydrogel structure is determined. The fabricated polymeric cross-linked hydrogel patches were dried and weighed (*m_c_*). They were then immersed in distilled water for a week with occasional shaking. This extraction procedure easily removes the polymer contents that are uncross-linked from the gelled hydrogel. Patches were then taken out from the solution, positioned in previously labeled petri dishes. They were then placed in a vacuum oven, maintained at 60 °C. Patches were dried until constant weight was obtained, which was termed as (*m_d_*). The equation given below was used for determining *Sol* % and the *Gel* % [17].
(2)$$Gel\;\%=\frac{m_{d}}{m_{c}}\times 100$$
(3)Sol %=100−Gel %

### 3.8. Drug-Loading Study of Formulation

Bacitracin zinc was loaded in formulated polymeric hydrogel topical patches. An amount of 1% of the drug was mixed in the 0.1 M HCL solution until a clear solution was obtained. The developed dried hydrogel patches were first weighed and then soaked in drug solution for 24 h at room temperature. The drug-loaded hydrogel topical patches were taken out from the drug solution after the above-mentioned specific period of time and washed with distilled water to remove the excessive solution contents from the surface of the patch. They were then dried in vacuum oven at 40 °C and weighed again [18]. The amount of drug being loaded in hydrogel patches was determined by the following equation.
(4)$$\;\mathrm{Percent}\;\mathrm{drug}\;\mathrm{loading}=\frac{Final\;weight\;of\;hydrogel\;patch-Initial\;weight\;of\;hydrogel\;patch\;}{Total\;drug\;used\;in\;loading}\times 100$$
(5)$$\mathrm{Entrapment}\;\mathrm{Efficiency}\;\left(\%\right)=\frac{\mathrm{Actual}\;\mathrm{drug}\;\mathrm{contents}\;\mathrm{in}\;\mathrm{hydrogel}\;\mathrm{patch}}{\mathrm{Theoretical}\;\mathrm{drug}\;\mathrm{contents}\;\mathrm{in}\;\mathrm{hydrogel}\;\mathrm{patch}}\times 100$$

### 3.9. In Vitro Drug Release Study of Topical Patch

The in vitro release pattern of bacitracin zinc-loaded cross-polymeric hydrogel topical patches was observed at different pH levels, ranging from normal skin pH, that is, 5.5, to infected skin pH, that is, 6.5 and 7.4. A USP type-Ⅱ dissolution apparatus was used for the evaluation of the controlled drug-release pattern of loaded patches. Phosphate buffer solution of respective pH was taken up to 500 mL in the buckets of the dissolution apparatus maintained at 37 °C ± 0.5 and stirred at 50 rpm paddle speed. Drug-loaded topical patches were weighed and then immersed in the buffer-containing buckets. Samples were withdrawn with the help of a graduated pipette from the buckets at regular intervals of time with the replacement of fresh dissolution medium [18]. Samples were filtered, diluted with fresh buffer solution and then analyzed for drug release at 251 nm wavelength using a UV–VIS spectrophotometer.

### 3.10. Release Kinetic Modeling of Bacitracin Zinc

In order to determine the drug release behavior from the formulated polymeric cross-linked hydrogel topical patch, different kinetic models were applied, as the release pattern of the drug from a zero-order kinetic model is not reliant on the concentration of active therapeutic agent in the hydrogel topical patch.
(6)$$\;\;Q_{t}=Q_{0}+K_{0}t$$ where *Q_t_* = amount of drug dissolved within “*t*” time, *Q*_0_ = initial amount of drug in the solution and *K*_0_ = zero order release rate constant. In the first-order kinetic model, release is reliant on the concentration of therapeutically active ingredient in the hydrogel topical patch.
(7)$$\log C=\log C_{0}-\frac{Kt}{2.303}$$
where *C*_0_ = initial concentration of drug within patch, *K* = first order rate constant, *t* = time, and −*K*/2.303 = slope.

Higuchi’s model demonstrates that the release of the drug from the formulation follows Fick’s law. The following equation explains the mechanism:(8)$$F_{t}=K_{2}\;\times \;t^{1/2}$$
where *F_t_* = amount of undissolved drug, and *K*_2_ = rate constant.

The drug release pattern in Korsmeyer–Peppas model is expressed in the following equation:(9)$$M_{t}/M_{\infty }=Kt^{n}$$
where $M_{t}$/$M_{\infty }$ = fraction of drug release from polymeric system at “*t*” time, *K* = release rate constant, and *n* = diffusion exponent [19].

### 3.11. Primary Skin Irritation Study of Topical Patch

A skin irritation study was performed on albino rabbits using the Draize scale as standard to evaluate the skin irritation potential towards the applied topical patch [20,21]. Protocols for the study were permitted by the PREC (Pharmacy Research Ethics Committee) of the University of Lahore. White healthy albino male rabbits weighing 2–3 kg were taken and divided into three groups. Rabbit’s skin was made free from hair and proper care was taken to avoid any damage to skin layers during shaving. Group-I was tagged as control and was kept without any treatment. Group-II received marketed formulation, whereas formulated topical patches (without drug) were applied to Group-III. Patches were applied at area not exceeding 4 cm^2^. The experiment was performed for 3 days, and the application sites were graded according to the visual scoring scale. Signs of erythema and edema were scored as follows: 0 for none, 1 for slight, 2 for well-defined, 3 for moderate, and 4 for formation of severe scar [22].

### 3.12. In Vitro Drug Deposition Study through Semipermeable Synthetic Membrane

A Franz diffusion cell was used to study the drug deposition profile through the semipermeable synthetic membrane. Three pH values of 5.5, 6.5, and 7.4 were used to evaluate the in vivo performance of drug-loaded topical patches. In the diffusion cell, phosphate buffer of respective pH was thermostated at 32 ± 0.5 °C with continuous stirring in the receptor compartment during the whole experiment. The cellophane membrane (0.45 µm) was mounted between the donor and the receptor compartment, a sample of formulated hydrogel matrix topical patch with an area of 1.5 cm^2^ was applied on the donor compartment over the synthetic membrane. Samples of 1 mL were withdrawn from the diffusion cell at specific time intervals with the replacement of fresh medium to the cell. Samples were diluted with the medium and analyzed by UV-Spectrophotometer at 251 nm.

### 3.13. Wound Healing Performance of Topical Patch

This study was performed to compare the wound healing potential of the fabricated polymeric cross-linked hydrogel topical patch with that of market formulation. Healthy white albino rabbits were selected for this activity. Rabbits were divided into 3 groups. Group-1 named as control group received no treatment. Group-2 received marketed formulation, whereas fabricated cross-polymeric topical patches containing the model drug were applied to the Group-3. Hairs from the abdominal area were removed prior to the creation of wound. Next day superficial wound was created under local anesthesia. All animals were kept in separate cages. The control group was left undressed, Group-2 and Group-3 were treated with marketed formulation and fabricated topical formulation. Rabbits were kept under observation until healing was observed [23].

## 4. Results and Discussion

### 4.1. Physical Appearance

Polymeric cross-linked hydrogel topical patches were synthesized by a free radical polymerization procedure using different concentrations of polymer along with varied concentrations of monomer and cross-linking agent. The physically stable patches obtained were transparent in color, soft, adhesive, elastic in nature with good mechanical and gelling strength. The physical appearance of the fabricated polymeric cross-linked hydrogel topical patch is shown in Figure 1.

### 4.2. Fourier-Transform Infrared Spectroscopy (FTIR Spectroscopy)

FTIR spectroscopy of PEG-8000, AA, C934, Tween 80, and MBA was performed to analyze the chemical and structural relationship of the components of the polymeric topical patches. The PEG-8000 spectrum showed a strong and broad absorption band at 3470 cm^−1^ due to O-H and C-OH stretching vibrations. Peak stretching at 2880 cm^−1^ was due to the aliphatic C-H group and vibrational CH₂. Bending vibrations at 1466 cm^−1^ and 1369 cm^−1^ were due to the C-H bond. Bands at 1146 cm^−1^ and 1094 cm^−1^ were due to stretching vibrations of O-H and C-O-H bonds [24]. The FTIR spectra of acrylic acid (Figure 2c) endorsed the occurrence of vibrational stretching of CH₂ at 2970 cm^−1^. At 1630 cm^−1^ and 1290 cm^−1^, C=C and C-O also showed vibrational stretching. A peak at 1709 cm^−1^ represented the occurrence of carboxylic acid as a functional group in the structure of acrylic acid [25]. The FTIR spectrum of C934 indicated peak at 2950 cm^−1^ due to the stretching vibrations of the OH group. A very sharp peak at 1700 cm^−1^ indicated the prevalence of carbonyl C=O stretching. A peak at 1450 cm^−1^ was assigned to C-O and O-H group stretching, whereas the band observed at 1250 cm^−1^ represented stretching of the C-O-C bridge of acrylates. Out-of-plane bending of the C=CH group was also observed at 850 cm^−1^ [26]. The spectrum of bacitracin zinc showed an absorption band at 3296 cm^−1^, representing stretching vibrations of a N-H group of secondary amines. A sharp peak at 1650 cm^−1^ confirmed the occurrence of a C=O group. The same peak at 1650 cm^−1^ also predicted the existence of a C=N group just like the C=O group. A band at 1535 cm^−1^ indicated the presence of aromatic C=C stretching [27]. The spectrum of MBA showed a stretching band at 3303 cm^−1^ due to the stretching of the N-H group, whereas a sharp peak at 1650 cm^−1^ appeared due to the stretching of its characteristic carboxylic group. Bands appearing at 3102 cm^−^¹ and 3032 cm^−^¹ were due to the symmetrical and asymmetrical stretching of the CH_2_ group in its structure. These bands were also observed by Reddy and his co-workers [28].

FTIR spectra of the unloaded topical patch showed prominent stretching of peaks of PEG-8000 at 2880 cm^−^¹ and 2900 cm^−^¹ due to aliphatic C-H group and vibrational movements of CH_2_, at 1700 cm^−^¹, indicating the stretching of the carbonyl C=O group as well as the C=N group, similar to C=O, between 1190–1200, representing the stretching of the C-O-C bridge. The drug-loaded (bacitracin zinc) formulation showed a characteristic band at 3200 cm^−1^ representing the stretching of the N-H group, whereas another peak at 1650 cm^−1^ was related to the incorporation of the drug in the formulation [29]. The FTIR spectra of all contents are given in Figure 2.

### 4.3. Thermogravimetric Analysis and Differential Scanning Calorimetry (TGA and DSC)

Thermal analysis of the raw polymer and optimized hydrogel topical patch was performed. The TGA of PEG-8000 revealed major weight loss at 360 °C to 420 °C, corresponding to the thermal decomposition of PEG functional groups [24]. Obtained results show that the polymer is highly unstable at elevated temperature, as abrupt degradation occurred at elevated temperatures. Whereas the major degradation of PEGAAM formulation occurred between 220 °C to 370 °C due to the breakage of main polymer chains, final degradation occurred at 460 °C to 480 °C. Results have shown that the complete thermal degradation of the PEGAAM formulation was delayed, as compared to PEG-8000. The immediate weight loss was also delayed in PEGAAM.

The raw polymer and PEGAAM were also analyzed for their thermal behaviors towards heat. Their thermodynamic properties were evaluated through DSC. A DSC thermogram of PEG-8000 revealed that the endothermic peak observed at 72.5 °C was due to the loss of moisture of the hydrophilic groups in the polymer, while complete degradation of the polymer was observed at 415 °C. DSC of PEGAAM showed two endothermic peaks. The first endothermic peak was observed at 110 °C, while the second peak was observed at 290 °C. The first peak corresponded to the moisture loss from the formulation, while second peak was due to the degradation of the polymeriche following equation explains the network in the formulation. The exothermic peak was seen around 430 °C, which showed that the peaks were shifted forward in PEGAAM as compared to the raw polymer, showing more stability of the formulation towards heat. The TGA and DSC study of individual reaction content and optimized formulation are mentioned in Figure 3.

### 4.4. Scanning Electron Microscopy (SEM)

The surface morphology of the hydrogel topical formulation was performed for the surface texture and insight surface analysis. SEM micrographs showed a visible rough and porous structure, as shown in Figure 4. The obtained porous structure is suitable for the penetration of water into the hydrogels, causing the formulation to swell. Obtained results were quite favorable for excellent drug entrapment into the formulation. After contacting the body fluids, the formulation showed good drug release.

### 4.5. Powder X-ray Diffraction (PXRD)

PXRD was performed to differentiate the nature of the formulation as well as that of the polymer. The crystalline and amorphous nature has a great impact on the solubility and release pattern from the formulation. PXRD of PEG-8000 showed sharp peaks at 23 θ and 18 θ [30]. These results signify the crystalline nature of the polymer, as shown in Figure 5a, whereas in formulation PXRD analysis (Figure 5b), the sharp and prominent peaks of the polymer were replaced by low intensity and dense peaks, representing the amorphous nature of the polymeric hydrogel formulation.

### 4.6. Effect of the Polymer, Monomer, and Cross-Linker on Swelling

Polymeric cross-linked hydrogel topical patches were prepared using varying concentrations of polymer (PEG-8000), monomer (AA), and cross-linker (MBA). These patches were analyzed for their swelling behavior in different buffer solutions of pH 5.5, 6.5, and 7.4. Swelling and drug release are directly linked with the hydrophilicity of the polymeric networks; therefore, swelling behavior was analyzed until a constant weight was obtained. PEG-8000 contains higher number of -OH and -COOH groups that impart a pH-sensitive character to this polymer and thus making it more attractive towards high pH media. Formulations from PEGAAM-1 to PEGAAM-3 have increasing concentration of polymer and the results have shown that with the increasing concentration of polymer, more swelling was observed. With the increase in the number of PEG-8000 molecules, the ionization and electrostatic repulsion of functional groups are expected to be increased, resulting in more equilibrium water imbibition of hydrogels. All formulated patches have shown higher swelling at pH 7.4 as compared to 6.5 and 5.5. At higher pH levels, the polymeric chain becomes ionized and enhances the water up-taking efficiency, resulting in increased swelling [31,32].

The formulation series from PEGAAM-4 to PEGAAM-6 contains increasing concentrations of AA. More swelling was observed in PEGAAM-6 because as the concentration of AA increases, the number of carboxylic groups also increases, which is responsible for the swelling of the hydrogel polymeric networks and at higher pH levels. The carboxylic group ionizes, leading to higher concentration of negatively charged COOˉ groups, thus resulting in increased electrostatic repulsion and a sharp increase in osmotic pressure in the hydrogel structure, leading to further swelling and expansion of the cross-linked polymeric network [33,34,35]. The formulation series from PEGAAM-7 to PEGAAM-9 contained increasing concentrations of MBA and showed less swelling. Swelling gradually decreased due to the reduction in spaces between the polymeric networks, thus, the denser structure restricted the hydrogel chain relaxation and swelling [36]. The swelling dynamics in time “t” (Qt) at different pH levels of an optimized formulation is shown in Figure 6.

### 4.7. Sol-Gel Analysis

In the formulation series from PEGAAM-1 to PEGAAM-3, the gel fraction improved as the quantity of polymer was increased. PEGAAM-3 showed the highest gel fraction of 99.3%. In the formulation series from PEGAAM-4 to PEGAAM-6, the concentration of monomer (AA) was increased while keeping the polymer concentration constant. Results have shown a gradual increment in the gel fraction with increasing monomer concentration. This is because by increasing the concentration of both polymer and monomer, the available active sites and functional groups increase for free radical polymerization, leading to increased gel fraction, and a more stable hydrogel is formed [37]. Moreover, the formulation series from PEGAAM-7 to PEGAAM-9 showed an increased gel fraction from 89.5% to 99.3% with increasing MBA concentration. The sol fraction was also examined to calculate the amount of unreacted reactants (polymer, monomer cross-linker) in the synthesis of polymeric hydrogel topical patches. Results showed minimal sol fraction, indicating the successful formation of cross-linked polymeric hydrogel networks in the topical patches. The gel fraction studies on individual reaction components are mentioned in Figure 7.

### 4.8. Influence of Reactants on Gel %, Yield % and Gel Time of Hydrogel Topical Patch

In the formulation series from PEGAAM-1 to PEGAAM-3, gel % and yield % were increased with the increase in polymer (PEG-8000) concentration, whereas a decrease in gel time was observed. The formulation series from PEGAAM-4 to PEGAAM-6 showed a similar pattern of increasing gel % and yield %, with the increase in the monomer (AA) concentration and decreased gelling time. This is because, with the increase in the concentration of polymer and monomer, available active sites for free radical polymerization increases, causing early entanglement of polymeric chains, accelerating the process of gel formation. Decreased gel time and increased gel % and yield % were also observed with increasing MBA concentration. The cross-linker caused an increased reaction rate, leading to the formation of polymeric chains at a much faster rate, leading to increased gel % and yield % [38]. The yield %, gel %, and gel time of PEG-8000, AA, and MBA, respectively, are shown in Figure 8. Increase gel/yield % indicates successful development of the formulations. More than 80% of the gel/yield % was observed for all the formulations, indicating successful formulation of the topical patch.

### 4.9. Drug-Loaded Content and Drug Entrapment Efficiency (%)

Drug entrapment efficiency in the formulation series from PEGAAM-1 to PEGAAM-9 were evaluated and the results obtained show (Table 2) that the formulation having the maximum concentration of polymer showed significant results for the percent of drug entrapment efficiency.

### 4.10. In Vitro Drug Release Study

Polymeric cross-linked hydrogel topical patches were evaluated for their drug release behavior in media of different pH levels, i.e., 5.5, 6.5, and 7.4. These selected pH ranges were in accordance with the pH of normal to infected skin. Obtained results showed an increase in the drug release pattern from formulations PEGAAM-1 to PEGAAM-3, due to increasing polymer concentration. PEGAAM-3 showed the maximum drug release at pH 7.4. At higher pH levels, hydrogen bonding between the OH^−^¹ of PEG-8000 decreases due to increased electrostatic repulsion which leads to widening of the mesh of the polymeric network. Thus, more water molecules enter into and out of the polymeric network, explaining the increased drug release from the formulation [39]. Formulations from PEGAAM-4 to PEGAAM-6 showed increased drug release with gradual increase in the monomer concentration. Similarly, at pH 7.4, more flux of drug through the PEGAAM-6 was observed than at pH 6.5 and 5.5. After a 6 h time interval, the optimized formulation PEGAAM-6 showed 16.562% drug release at pH 5.5, 21.384% at pH 6.5, while a total of 25.996% drug release was found at pH 7.4. Similarly, after a duration of 12 h, a total of 20.545%, 27.463%, and 35.430% drug release was found in pH 5.5, 6.5, and 7.4. After an interval of 48 h, the drug release amount in pH 5.5, 6.5, and 7.4 was 38.784%, 69.182%, and 81.971%. At the end of the study, after a duration of 72 h, the drug release amount at pH 5.5, 6.5, and 7.4 was 54.507%, 85.954%, and 97.694%. Formulations from PEGAAM-7 to PEGAAM-9 showed a decline in drug release. This is because these formulations contain increasing cross-linker concentrations. With the increases in the concentration of cross-linker, hydrogen bonding between-OH groups become stronger, which leads to a reduction in electrostatic repulsive forces, which makes the polymeric network stronger and denser. Thus, drug loading and release were restricted with increasing MBA concentration. The drug release percent of an optimized formulation (PEGAAM-6) is elaborated in Table 3.

### 4.11. In Vitro Drug Deposition Study through Semipermeable Synthetic Membrane

The results of the study have shown that the amount of model drug (bacitracin zinc) permeated through the semipermeable synthetic membrane of area 1.5 cm^2^ is more at pH 7.4 than at 6.5 and 5.5. Permeability flux (Jss) and permeability coefficient (Kp) are mentioned in Table 4. Results obtained showed that the permeation flux of the drug-loaded topical patch is more at pH 7.4 (1.35551 µg/cm^2^/h), whereas it is lower at pH 5.5 (0.7274 µg/cm^2^/h). Results also showed that the permeability coefficient (Kp) is also 2.711 × 10^−6^ cm/h at 7.4, whereas it is lower at pH 5.5. The kinetic analysis applied on the obtained permeation data against time represented a zero-order release of the drug from the patch across the skin. Drug permeation from the patch followed a diffusion-controlled mechanism. Results indicated that the drug permeation is more at pH 7.4, which created an ideal drug residence at the infected skin surface. Hence, improvement in the in vitro availability of the drug through the formulation across the skin was observed. Drug permeation across the synthetic membrane of the optimized formulation with time at different pH is given in Figure 9.

### 4.12. Kinetic Modeling

The (bacitracin zinc) drug-release kinetics of polymeric cross-linked hydrogel topical patches were analyzed by applying different kinetic models, such as first-order, zero-order, Higuchi, and Korsmeyer–Peppas models, shown in Table 5. Drug-release data and DD solver software were used for the determination of the release pattern from the designed drug delivery system. All the nine formulations followed zero-order kinetics, having regression co-efficients (R^2^) in the range of 0.9791–0.9960, and the Higuchi model, with regression co-efficients (R^2^) in the range of 0.9105–0.9680. It is evident from our results that the drug release from the hydrogel matrix of the topical patch was independent of the drug concentration and followed a mechanism of diffusion by forming pores in the polymeric matrix. Therefore, our formulation contained a suitable porous structure for water imbibition and diffusion of the drug from the matrix. The “*n*” exponent value ranged between 0.488 and 0.628.

### 4.13. Primary Skin Irritation Study of Topical Patch

A primary skin irritation study was performed on healthy white albino rabbits to evaluate the irritation potential of the formulated topical patch towards their intact skin. Optimized hydrogel topical patches were applied on the intact skin of the healthy rabbits. The evaluation of the test was carried out according to the protocols provided by the Pharmacy Research Ethics Committee (PREC) of the University of Lahore. The application site was observed at 24, 48, and 72 h. No signs of erythema and irritation were seen on the rabbit’s skin over a period of 72 h. Therefore, the components of the topical patch are safe for topical delivery. Obtained results favored the skin acceptance towards the formulated hydrogel topical patch. The evaluation of the rabbit skin for irritation is shown in Figure 10. Moreover, the scores of the skin irritation study of an optimized formulation are mentioned in Table 6.

### 4.14. Wound Healing Performance of Topical Patch

This study was performed to study the wound healing potential of the fabricated topical patch, comparing it to the market formulation shown in Figure 11. Healthy white albino rabbits were selected for this activity. Rabbits were divided into three groups. Group-1 was named as the control group, which received no treatment. Group-2 received a marketed formulation, whereas the fabricated cross-polymeric topical patch containing the model drug was applied to the Group-3. All animals were kept in separate cages. Wound healing of rabbits was examined for 72 h. Results showed that the wound healing potential of the formulated hydrogel topical patch containing the model drug (bacitracin zinc) was higher as compared to the marketed formulation. Complete healing of the wound was observed in group-3 over a period of 72 h, whereas small rashes were seen on the wounds of group-2 animals.

## 5. Conclusions

It is concluded that polymeric cross-linked hydrogel topical patches were synthesized successfully by free radical polymerization. Fabricated cross-linked topical patches were evaluated for their appearance, morphology, thermodynamic stability, crystallinity, swelling, sol-gel fraction, and drug-release pattern from the polymeric matrix. Fabricated topical patches were also tested for their percentage of drug permeation across skin, along with its in vitro release through the semisynthetic membrane, sensitivity, and healing performance. Characterization studies confirmed the formation of a new, stable polymeric cross-linked hydrogel network. The formulation containing the highest polymer, monomer and lowest cross-linker concentrations showed the highest swelling and percentage drug release, along with higher concentrations of drug permeating across the skin with zero-order release kinetics. All formulations showed pH-sensitive behavior. The drug-release and drug-permeation studies showed excellent results at elevated pH (7.4), indicating its suitability for the wounded skin surface. Fabricated patches showed no signs of sensitivity or erythema after their application on skin for 72 h. Moreover, healing performance was found to be faster with the fabricated patch than the marketed formulation. All these results have shown that these cross-linked hydrogels topical patch are excellent candidates for the controlled release of drugs on the wounded skin surface locally and systemically for wound healing and controlling wound infection.

## Figures and Tables

**Figure 1 polymers-15-01652-f001:**
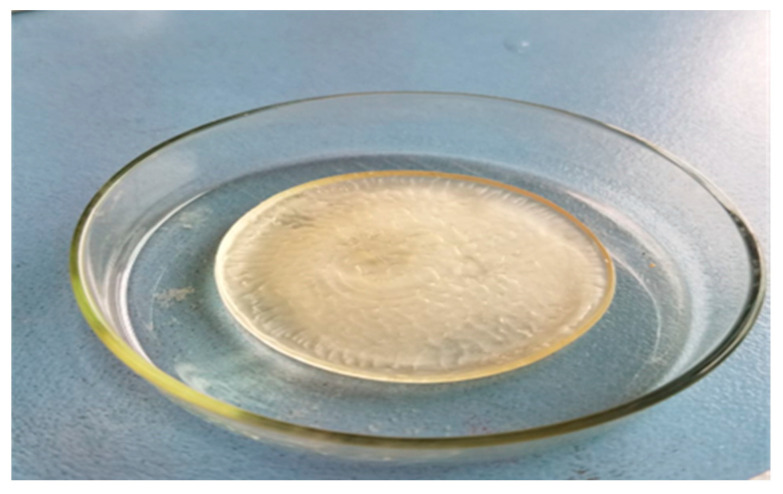
Fabricated polymeric cross-linked hydrogel topical patch.

**Figure 2 polymers-15-01652-f002:**
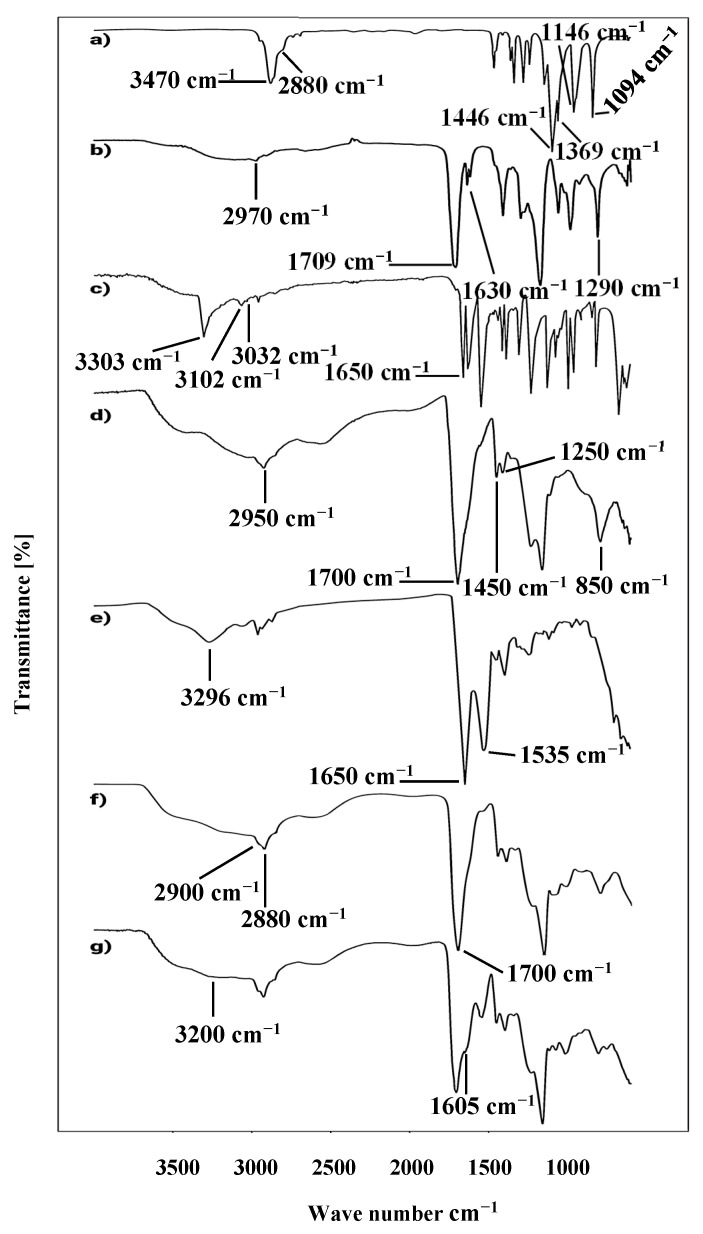
FTIR spectrum of (**a**) PEG-8000, (**b**) acrylic acid, (**c**) MBA, (**d**) carbopol 934, (**e**) bacitracin zinc, (**f**) unloaded formulation, and (**g**) drug-loaded formulation.

**Figure 3 polymers-15-01652-f003:**
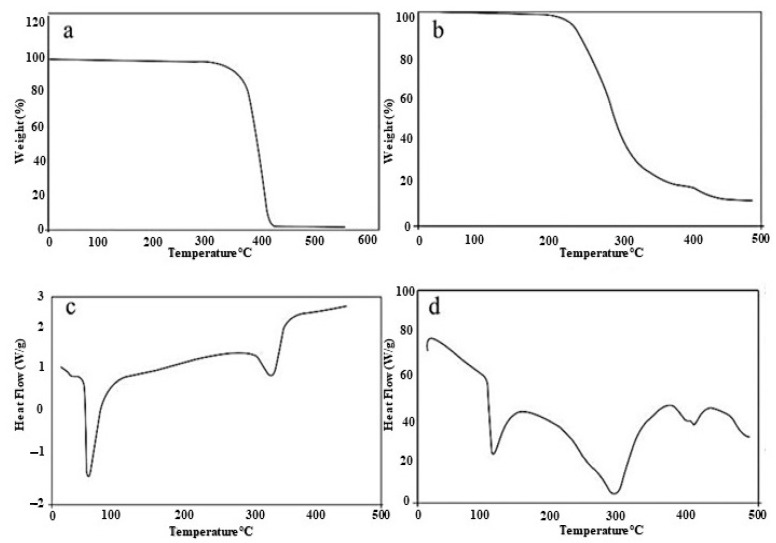
(**a**) TGA of PEG-8000. (**b**) TGA of formulation. (**c**) DSC of PEG-8000. (**d**) DSC of formulation.

**Figure 4 polymers-15-01652-f004:**
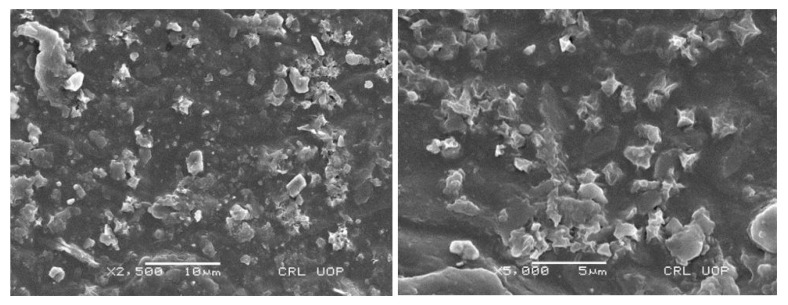
Surface morphology of PEGAAM formulation at different magnifications.

**Figure 5 polymers-15-01652-f005:**
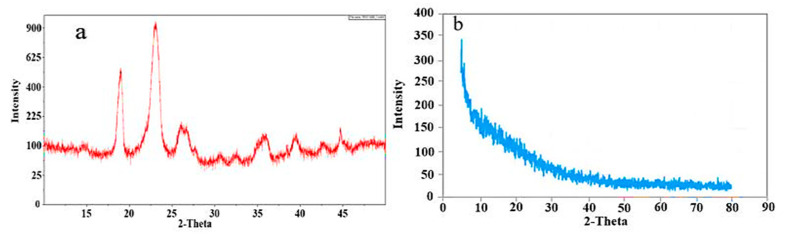
PXRD of (**a**) PEG-8000 and (**b**) formulation.

**Figure 6 polymers-15-01652-f006:**
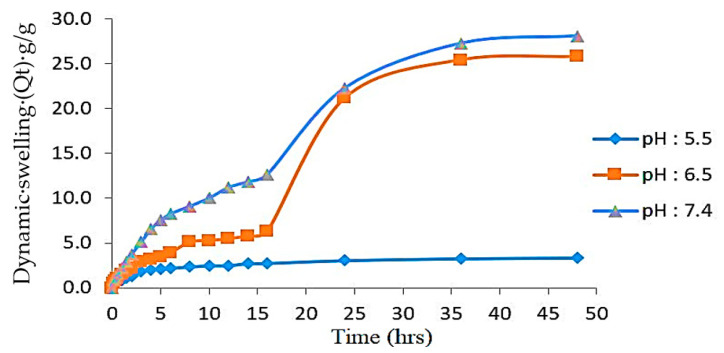
PEGAAM swelling dynamics at different pH levels.

**Figure 7 polymers-15-01652-f007:**
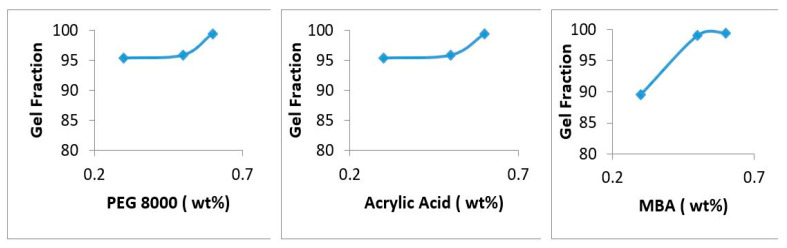
Gel fraction of PEG-8000, acrylic acid (AA), and MBA.

**Figure 8 polymers-15-01652-f008:**
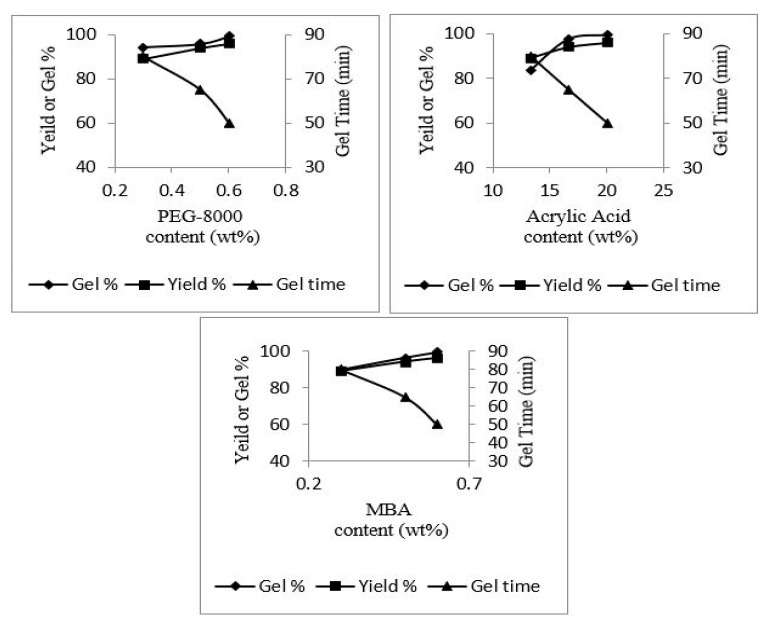
Yield %, gel %, and gel time of PEG-8000, AA, and MBA.

**Figure 9 polymers-15-01652-f009:**
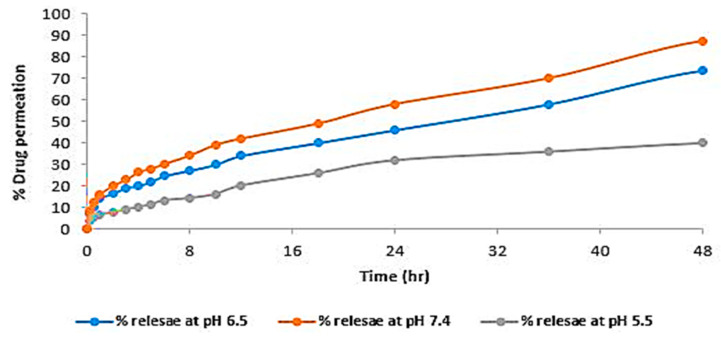
Drug permeation across synthetic membrane with time at different pH.

**Figure 10 polymers-15-01652-f010:**
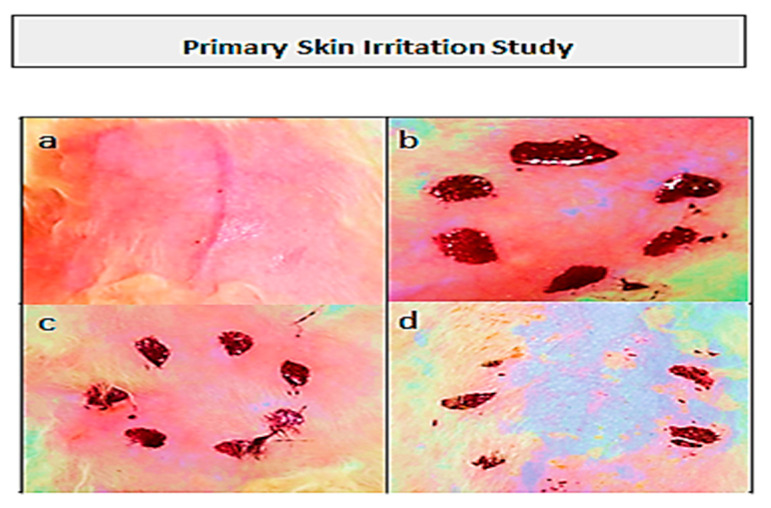
Evaluation of rabbit skin for irritation study (**a**) fresh shaved skin (**b**) after 24 h (**c**) after 48 h, (**d**) after 72 h.

**Figure 11 polymers-15-01652-f011:**
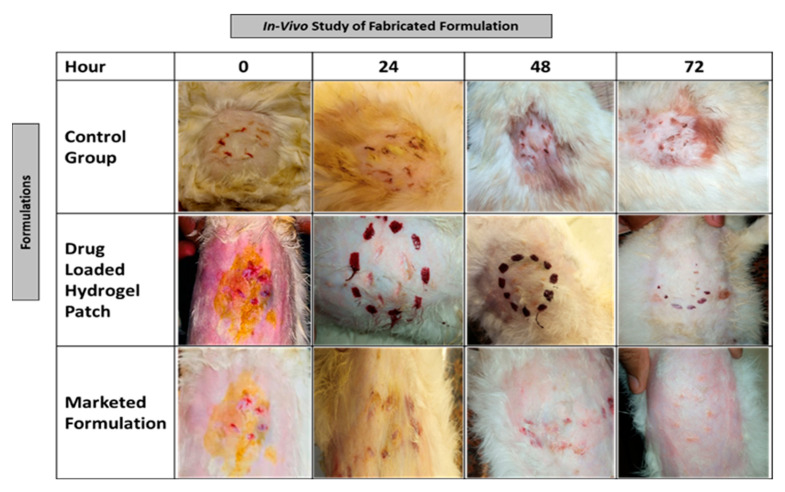
Healing performance of optimized drug-loaded formulation with control group and marketed formulation.

**Table 1 polymers-15-01652-t001:** Composition of fabricated PEGAAM formulations.

Scheme 8000.	Formulation Code	PEG-8000 (g)	Carbapol-934 (g)	Tween 80 (wt. %)	AA (g)	APS (g)	MBA (g)
1	PEGAAM-1	0.1	0.01	0.01	6	0.08	0.1
2	PEGAAM-2	0.15	0.01	0.01	6	0.08	0.1
3	PEGAAM-3	0.2	0.01	0.01	6	0.08	0.1
4	PEGAAM-4	0.2	0.01	0.01	4	0.08	0.1
5	PEGAAM-5	0.2	0.01	0.01	5	0.08	0.1
6	PEGAAM-6	0.2	0.01	0.01	6	0.08	0.1
7	PEGAAM-7	0.2	0.01	0.01	6	0.08	0.1
8	PEGAAM-8	0.2	0.01	0.01	6	0.08	0.15
9	PEGAAM-9	0.2	0.01	0.01	6	0.08	0.2

**Table 2 polymers-15-01652-t002:** Drug entrapment efficiency.

Sr. No.	Formulation Code	Drug Loading (%)	Drug Entrapment Efficiency (%)
1	PEGAAM-1	83	68
2	PEGAAM-2	85	70
3	PEGAAM-3	88	75
4	PEGAAM-4	80	66
5	PEGAAM-5	81	67
6	PEGAAM-6	86	72
7	PEGAAM-7	87	73
8	PEGAAM-8	78	64
9	PEGAAM-9	76	62

**Table 3 polymers-15-01652-t003:** Drug release of the optimized formulation PEGAAM-6.

Time (h)	Percent Drug Release (pH 5.5)	Percent Drug Release (pH 6.5)	Percent Drug Release (pH 7.4)
0.33	10.482	13.417	16.771
0.5	10.901	14.465	17.400
1	11.321	15.303	18.029
2	12.788	16.981	19.077
3	13.836	17.191	20.335
4	14.465	18.239	22.222
5	14.884	19.077	23.270
6	16.561	21.383	25.995
8	17.400	23.270	27.253
10	18.658	25.786	31.446
12	20.545	27.463	35.429
16	22.641	32.285	38.155
18	24.528	36.477	42.347
24	28.721	45.702	55.555
36	31.656	56.603	71.069
48	38.784	69.182	81.970
60	49.475	75.471	92.033
72	54.507	85.953	97.693

**Table 4 polymers-15-01652-t004:** Permeability flux (Jss) and permeability coefficient (Kp) at different pH.

Release of Drug at Different pH	Jss (µg/cm^2^/h)	Kp (cm/h)
5.5	0.7274	1.454 × 10^−6^
6.5	1.0767	2.153 × 10^−6^
7.4	1.3555	2.711 × 10^−6^

**Table 5 polymers-15-01652-t005:** Kinetic modeling on bacitracin zinc-loaded topical patch.

Formulations	Zero-Order R^2^	First-Order R^2^	Higuchi Model R^2^	Krosmeyer–Peppas R^2^	*n*
PEGAAM-1	0.9960	0.9636	0.9105	0.8463	0.542
PEGAAM-2	0.9914	0.9641	0.9448	0.8890	0.553
PEGAAM-3	0.9947	0.9668	0.9657	0.8665	0.522
PEGAAM-4	0.9822	0.9306	0.9026	0.8874	0.551
PEGAAM-5	0.9963	0.9696	0.9564	0.9240	0.628
PEGAAM-6	0.9947	0.9668	0.9657	0.8665	0.522
PEGAAM-7	0.9943	0.9669	0.9658	0.8667	0.522
PEGAAM-8	0.9850	0.9745	0.9507	0.9397	0.557
PEGAAM-9	0.9791	0.9675	0.9680	0.9345	0.488

**Table 6 polymers-15-01652-t006:** Scores of Skin Irritation Study (Mean ± SE) of optimized formulation.

Formulations	Irritation Score (*n* = 3)		
	Time of Application		
	24 h	48 h	72 h
Control	0 ± 0	0 ± 0	0 ± 0
Marketed formulation	0 ± 0	0 ± 0	0 ± 0
Hydrogel Topical Patch	0 ± 0	0 ± 0	0 ± 0

Draize Patch irritation scale: 0 for none; 1 for very slight; 2 for well-defined; 3 for moderate and 4 for eschar formation.

## Data Availability

Not applicable.

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
