# Peer review of "Formulation and Characterization of Polymeric Cross-Linked Hydrogel Patches for Topical Delivery of Antibiotic for Healing Wound Infections"

_polymers, 2023, doi:10.3390/polym15071652_

Round 1
Reviewer 1 Report
The manuscript is well-described and contains interesting information. I have found certain issues while reviewing the manuscript and recorded them below.
1. Introduction/ para-1/ last line: “Wound exudate fails--- ---“ is not meaningful. Please rewrite the same.
2. Para-2 speaks about the importance of hydrogels, but before, the authors should discuss the limitations of the conventional treatment and the requirements of hydrogels. Then, the continuity will be there.
3. Introduction/ last but one para/ line no. 84: “Marketed formulations… ….”. Please rewrite the same. It looks like the conclusion of the present study. It will be better if authors state the limitations of marketed formulations (as stated) and subsequent requirement of alternative formulations that provide controlled release etc.
4. Introduction/ last para: “ structural analysis --- ---performed” – please delete this portion, and the 1st line (The purpose of this study--- ---) should be added to the end of the previous paragraph.
5. 4.3: t will be Fourier Transform Infrared Spectroscopy. Please delete ‘s’ from transforms.
6. 2.1/ last line: it will be ‘is shown’ instead of ‘in mentioned.’
7. Figure-2/ wavenumber is a single word, and the unit should be written in the bracket.
8. 2.4: ‘After contacting the body fluids, the formulation has shown good drug release.” How can it be confirmed from SEM?
9. Figure 5: XRD pattern of PEG is not required, and it's just meaningless. XRD image of patches is important for this study. The 5B image is not clear, and it shows only the amorphous structure. The quality should be improved.
10. Figure 6: Y-axis represents Qt. What does it mean? What is its unit of it?
11. 4.6: Authors discussed the increased gel% and yield%, but they did not quantify it. Quantifications are required to provide clear information.
12. 4.7: label is for loading efficiency (LDC) and EE, but loading information is missing.
13. The release profile shows continuous increments even after 60h. What is the reason? What about the sustained release concepts?
14. Authors can provide information on the quantity of drug released after a proper interval.
15. Authors can improve their writing skills. There are several mistakes in the manuscript.
16. Decision: Revision.
Author Response
Dear Editor
We are grateful to the reviewers for sparing their precious time to review our paper. There suggestions were profoundly helpful to improve the overall quality of our paper. The Manuscript has been revised accordingly and the changes have been highlighted yellow in the paper.
Moreover, our point-by-point response to reviewers’ comments are given as follows.
RESPONSES TO REVIEWER 1
Comment 1. Introduction/ para-1/ last line: “Wound exudate fails--- ---“ is not meaningful. Please rewrite the same
RESPONSE: As per the direction of the expert reviewer, the suggested change has been made.
Comment 2. Para-2 speaks about the importance of hydrogels, but before, the authors should discuss the limitations of the conventional treatment and the requirements of hydrogels. Then, the continuity will be there.
RESPONSE: As per the direction of the expert reviewer, the suggested addition has been done.
Comment 3. Introduction/ last but one para/ line no. 84: “Marketed formulations… ….”. Please rewrite the same. It looks like the conclusion of the present study. It will be better if authors state the limitations of marketed formulations (as stated) and subsequent requirement of alternative formulations that provide controlled release etc.
RESPONSE: As per the direction of the expert reviewer, the suggested change has been made.
Comment 4. Introduction/ last para: “ structural analysis --- ---performed” – please delete this portion, and the 1st line (The purpose of this study--- ---) should be added to the end of the previous paragraph.
RESPONSE: As per the direction of the expert reviewer, the suggested change has been made.
Comment 5. 4.3: t will be Fourier Transform Infrared Spectroscopy. Please delete ‘s’ from transforms.
RESPONSE: As per the direction of the expert reviewer, the suggested change has been made.
Comment 6. 2.1/ last line: it will be ‘is shown’ instead of ‘in mentioned.
RESPONSE: As per the direction of the expert reviewer, the suggested change has been made.
Comment 7. Figure-2/ wavenumber is a single word, and the unit should be written in the bracket.
RESPONSE: As per the direction of the expert reviewer, the suggested change has been made.
Comment 8. 2.4: ‘After contacting the body fluids, the formulation has shown good drug release.” How can it be confirmed from SEM?
RESPONSE: Since the SEM images have shown porous morphology of the hydrogel patches and it has also been stated in the mentioned section that porous structure is quite suitable for significant encapsulation of the drug in its porous structure. The relation between the SEM images and that of the drug release can be explained in terms of porosity. Since, the images show porous structure, therefore, as soon as the dissolution medium comes in contact with the patches, the system will swell ad the drug being encapsulated in the porous structure will get enough penetration of the medium, resulting in the magniloquent release of the drug from the system.
Comment 9. Figure 5: XRD pattern of PEG is not required, and it's just meaningless. XRD image of patches is important for this study. The 5B image is not clear, and it shows only the amorphous structure. The quality should be improved.
RESPONSE: We have added the XRD pattern oof the PEG (polymer) just to compare the crystallinity of the polymer with that of the formulation. The XRD pattern of the polymer intense peaks which signifies its crystalline nature. In comparison with the XRD pattern of the synthesized formulation (5B), the intense peaks of the polymer have been suppressed and the peaks of the synthesized formulation (5B) confirms that an amorphous system has been formulated. The image 5B is shows the amorphous diffractogram of the synthesized formulation, and because of this, the image only shows the amorphous structure which was the basic aim of the study.
Comment 10. Figure 6: Y-axis represents Qt. What does it mean? What is its unit of it?
RESPONSE: Qt stands for swelling dynamics in time “t”. It shows the amount of swelling the synthesized formulation has shown and its unit is “g/g” which has also been mentioned in the Figure 6.
Comment 11. 4.6: Authors discussed the increased gel% and yield%, but they did not quantify it. Quantifications are required to provide clear information.
RESPONSE: As per direction of the competent reviewer, the suggested addition has been made in the manuscript.
Comment 12. 4.7: label is for loading efficiency (LDC) and EE, but loading information is missing.
RESPONSE: As per direction of the competent reviewer, the missing section has been added in the manuscript. It was due to typographic mistake. The heading has been modified and corrected.
Comment 13. The release profile shows continuous increments even after 60h. What is the reason? What about the sustained release concepts?
RESPONSE: The formulated hydrogel patches were designed for the release of the drug for up to 72 hours in order to have a programmed release of the drug for the maximum therapeutic concentration. Because of this, the release profile has shown some of the increments of the drug even after 60h.
Comment 14. Authors can provide information on the quantity of drug released after a proper interval.
RESPONSE: As per direction of the competent reviewer, the suggested addition has been made in the manuscript.
Comment 15. Authors can improve their writing skills. There are several mistakes in the manuscript.
RESPONSE: As per direction of the competent reviewer, the manuscript has been rephrased and the writing skill has been improved.

Reviewer 2 Report
The manuscript entitled "Formulation and Characterization of Polymeric Cross-linked Hydrogel Patches for Topical Delivery of Antibiotic for Healing Wound Infections ". Authors have reported fabricating polymeric cross-linked hydrogel topical patches using free radical polymerization. However, there are some points that need to be corrected. Therefore, recommended the publication of this paper after major revision.
1. The places of the e-mail addresses in the address section should be arranged.
2. In the first few sentences of the abstract section, introductory sentences about the study should be written. Then the experiment and characterization sections should be mentioned.
3. Add the short name of the chemical N, N'-Methylenebisacrylamide, in line 22 next to it in parentheses.
4. Write the long names of the PEG and MBA chemicals in line 28.
5. The title layout of the article is wrong. The title order of the articles should be in the form of an abstract, introduction, materials and methods, results and discussion, and conclusion. Please make the necessary corrections.
6. Line 46,185,186: Check the punctuation marks in the sentence.
7. Line 75: The name of bacteria in parentheses should be written in italics.
8. Line 89: The abbreviations of the techniques used should be written clearly form where they are mentioned for the first time in the text. Then abbreviations should be used throughout the text. Please check the whole article.
9. Line 105: The complete form of abbreviations is written only once in the first mention of the article. For example, PEG is mentioned many times, along with its full form.
10. A detailed explanation of the peak points of the materials is given in the text. In addition, it may be beneficial for the reader to indicate the peaks on the graph if technically possible.
11. Line 145: Zero in a sentence is meaningless.
12. There should be a certain writing form in the titles. In some titles, abbreviations of the analyzes are given, while long forms are written in others. Please make the necessary changes.
13. The image quality of figures 2,5,9,10, and 11 are very low. Please change the images.
14. How is the numbering in the PEGAAM formulation done? (PEGAAM-1,3,..) Please explain briefly in the text.
15. The explanations under the heading 4.8 in vitro drug release study are not clearly stated in figure 9. The comments made according to the increasing PEGAAM number are not reflected in the chart. A drug release table can be drawn specifically indicating PEGAAMs and varying acid ratios. In this way, the results are clearly stated to the reader.
16. line 327: please correct the typo.
17. What exactly is the name of the material that is said to be synthesized under the title of Fabrication of polymeric cross-linked hydrogel patches? PEGCAM or PEGAAM?
18. Please add title numbers to the methods described under "Characterization of fabricated polymeric cross-linked patches."
Author Response
The Editors
Polymers (Special issue: Polymer Materials for Drug Delivery and Tissue Engineering II)
Subject: Response to reviewers’ comments (Polymers-2171488)
Dear Editor
We are grateful to the reviewers for sparing their precious time to review our paper. There suggestions were profoundly helpful to improve the overall quality of our paper. The Manuscript has been revised accordingly and the changes have been highlighted yellow in the paper.
Moreover, our point-by-point response to reviewers’ comments are given as follows.
Dr. Kashif Barkat
Corresponding author/Associate Professor
Email: dr.kashif2009@gmail.com
Mobile: +92-300-9681946
RESPONSES TO REVIEWER 2
Comment 1. The places of the e-mail addresses in the address section should be arranged.
RESPONSE:
Comment 2. In the first few sentences of the abstract section, introductory sentences about the study should be written. Then the experiment and characterization sections should be mentioned.
RESPONSE: As per direction of the respected reviewer, the suggested changes have been made.
Comment 3. Add the short name of the chemical N, N'-Methylenebisacrylamide, in line 22 next to it in parentheses.
RESPONSE: As per direction of the respected reviewer, the suggested changes have been made.
Comment 4. Write the long names of the PEG and MBA chemicals in line 28.
RESPONSE: As per direction of the respected reviewer, the suggested changes have been made.
Comment 5. The title layout of the article is wrong. The title order of the articles should be in the form of an abstract, introduction, materials and methods, results and discussion, and conclusion. Please make the necessary corrections.
RESPONSE: As per suggestion of the expert reviewer, the title order has been corrected.
Comment 6. Line 46,185,186: Check the punctuation marks in the sentence.
RESPONSE: As per direction of the respected reviewer, the suggested changes have been made.
Comment 7. Line 75: The name of bacteria in parentheses should be written in italics
RESPONSE: As per suggestion of the expert reviewer, the suggested changes have been made.
Comment 8. Line 89: The abbreviations of the techniques used should be written clearly form where they are mentioned for the first time in the text. Then abbreviations should be used throughout the text. Please check the whole article.
RESPONSE: As per suggestion of the expert reviewer, the suggested changes have been made.
Comment 9. Line 105: The complete form of abbreviations is written only once in the first mention of the article. For example, PEG is mentioned many times, along with its full form.
RESPONSE: As per suggestion of the expert reviewer, the title order has been corrected.
Comment 10. A detailed explanation of the peak points of the materials is given in the text. In addition, it may be beneficial for the reader to indicate the peaks on the graph if technically possible.
RESPONSE: As per suggestion of the expert reviewer, the suggested changes have been made.
Comment 11. Line 145: Zero in a sentence is meaningless.
RESPONSE: As per suggestion of the expert reviewer, the suggested changes have been made.
Comment 12. There should be a certain writing form in the titles. In some titles, abbreviations of the analyzes are given, while long forms are written in others. Please make the necessary changes.
RESPONSE: As per suggestion of the expert reviewer, the suggested changes have been made.
Comment 13. The image quality of figures 2,5,9,10, and 11 are very low. Please change the images.
RESPONSE: As per suggestion of the expert reviewer, the quality of the images have been improved.
Comment 14. How is the numbering in the PEGAAM formulation done? (PEGAAM-1,3,..) Please explain briefly in the text.
RESPONSE: As per suggestion of the expert reviewer, explanation about the numbering has been done briefly in the manuscript.
Comment 15. The explanations under the heading 4.8 in vitro drug release study are not clearly stated in figure 9. The comments made according to the increasing PEGAAM number are not reflected in the chart. A drug release table can be drawn specifically indicating PEGAAMs and varying acid ratios. In this way, the results are clearly stated to the reader.
RESPONSE: Since the results were not clearly stated in Figure 9, therefore, the mentioned figure has been replaced with a table for more clarification and understanding of the results.
Comment 16. line 327: please correct the typo.
RESPONSE: As per suggestion of the expert reviewer, the suggested changes have been made.
Comment 17. What exactly is the name of the material that is said to be synthesized under the title of Fabrication of polymeric cross-linked hydrogel patches? PEGCAM or PEGAAM?
RESPONSE: It is PEGAAM and it has been corrected in the main manuscript.
Comment 18. Please add title numbers to the methods described under "Characterization of fabricated polymeric cross-linked patches."
RESPONSE: As per direction of the respected reviewer, numbering has been added.

Round 2
Reviewer 2 Report
The authors answered all of the comments.